# Knowledge, Attitudes, and Practices on COVID-19 Vaccination among General Adult Population in Malawi: A Countrywide Cross-Sectional Survey

**DOI:** 10.3390/vaccines12030221

**Published:** 2024-02-20

**Authors:** Master R. O. Chisale, Dzinkambani Kambalame, Saul Eric Mwale, Balwani Chingatichifwe Mbakaya, Regina Mankhamba, Pizga Kumwenda, Ben Chilima, Collins Mitambo, Mavuto Chiwaula, Billy Nyambalo, Clara Sambani, Jellita Gondwe, Charity Muwalo, Amon Dembo, Lines Chinyamunyamu, Mavuto Thomas, Matthews Kagoli, Evelyn Chitsa Banda

**Affiliations:** 1Biological Sciences Department, Faculty of Science, Technology and Innovations, Mzuzu University, Luwinga, Mzuzu Private Bag 201, Malawi; mwale.s@mzuni.ac.mw; 2Public Health Institute of Malawi, Ministry of Health, Lilongwe 00265, Malawi; dzinkambani@yahoo.com (D.K.); sekera2006@yahoo.co.uk (R.M.); chilimab2@gmail.com (B.C.); cmitambo@gmail.com (C.M.); mjchiwaula@yahoo.co.uk (M.C.); bnyambalo@gmail.com (B.N.); clara.sambani@kasungudc.gov.mw (C.S.); jellitagondwe@gmail.com (J.G.); lchinyamunyamu@gmail.com (L.C.); mkagoli@gmail.com (M.K.); evelyn.chitsabanda@health.gov.mw (E.C.B.); 3Public Health Department, University of Livingstonia, Mzuzu P.O. Box 112, Malawi; bcmbakaya@unilia.ac.mw; 4Biomedical Sciences Department, Faculty of Health, Mzuzu University, Luwinga, Mzuzu Private Bag 201, Malawi; kumwenda.p@mzuni.ac.mw; 5Health Education Unit, Ministry of Health, Lilongwe 00265, Malawi; charitygond@gmail.com (C.M.); amondembo@gmail.com (A.D.); mavutothomas@yahoo.co.uk (M.T.)

**Keywords:** COVID-19, vaccine, knowledge, attitude, practice, Malawi

## Abstract

Vaccination is one of the essential measures in reducing transmission, morbidity, and mortality rates of a disease. However, the COVID-19 vaccination is facing hesitancy across the globe, Malawi included. A population-based cross-sectional study was conducted in Malawi to document knowledge, attitudes, and practices on COVID-19 vaccination. The study targeted the general adult population and employed a multi-stage sampling technique. The Census Enumeration Areas within the 16 selected districts served as a primary sampling unit. Among the total 3068 participants, 1947 (63.6%) were female. About 1039 (34.1%) participants had primary education, while only 169 (5.5%) had college education. A total of 2936 (95.7%) participants knew about the COVID-19 vaccine, and 2063 (68.4%) felt that the COVID-19 vaccine was effective. A total of 1180 (38.7%) got vaccinated. Knowledge of the COVID-19 vaccination was significantly associated with participants’ education levels, location, occupation, marital status, household family income, and whether they were suffering from chronic illness or not. Overall, the level of knowledge and attitudes about the COVID-19 vaccination was good. This study has also established that different population groups have statistically different levels of knowledge and attitudes regarding COVID-19 vaccination. This study has also indicated a significant relationship between the rate of vaccination and several factors. Therefore, this calls for stakeholders to continue awareness and group-targeted tailored campaigns so as to increase COVID-19 vaccination.

## 1. Introduction

Coronavirus disease 2019 (COVID-19) is caused by severe acute respiratory syndrome coronavirus 2 (SARS-CoV-2), a highly pathogenic virus initially discovered in Wuhan, Hubei Province, China [1]. The World Health Organization has recognized COVID-19 as an ongoing pandemic that affects world security and causes economic instability [2].

In Malawi, the first case of COVID-19 was recorded on April 2, 2020. In the subsequent weeks, sporadic transmission clusters emerged in large cities like Blantyre, Lilongwe, Zomba, and Mzuzu. Additional importations of COVID-19 cases occurred among migrants returning primarily from South Africa, which was the African country with the largest documented outbreak at the time [3]. The recorded incidence of SARS-CoV-2 then increased in the subsequent months. Until July 5, 2023, Malawi had reported 86,600 confirmed cases of COVID-19, and approximately 2646 died from the disease, representing a mortality rate of 3.1%. This rate is higher as compared with the African rate of 1.8% and the global fatality rate of 1.2% [4]. This COVID-19 fatality rate in Malawi is also high if compared with neighboring countries like Zimbabwe (2.2%) and Zambia (1.3%) [4,5,6]. Likely, this is why Malawi appears to be in position 132 out of 230 countries currently reporting on COVID-19 [4,6]. However, when mortality is reported on a per 100,000 population basis, COVID-19 mortality in Malawi (14.04) was lower than that in Zambia (22.07) or Zimbabwe (38.16) [7].

Since the beginning of the pandemic, Malawi has experienced a total of four COVID-19 waves, with the second being the most severe. In response, the Malawi government instituted policies to promote the prevention of new infections. Among the preventative measures, vaccination was one of the key interventions adopted by the Ministry of Health when the vaccine became available. Johnson & Johnson, AstraZeneca, and Pfizer vaccines are some of the COVID-19 vaccines that have been approved and rolled out in Malawi. Studies have revealed that people have varying levels of knowledge, attitudes, and practices regarding vaccination, hence increasing the levels of COVID-19 hesitancy [8,9,10,11,12].

Few Knowledge, Attitudes, and Practices (KAP) studies conducted in Malawi have focused on COVID-19 preventive measures in selected districts. Despite Malawi rolling out a COVID-19 vaccination program for over a year, the vaccine uptake is very low. However, countrywide KAP studies to gauge why the vaccine uptake is low are limited. Such studies are crucial to identifying the factors contributing to low vaccine uptake and hesitancy.

Evidence suggests that public knowledge is vital in tackling pandemics [13]. Knowledge, attitudes, and practices (KAP) of the general population studies can provide useful insights to policymakers in designing proper interventions for controlling COVID-19. For instance, a KAP study conducted by Lee in South Korea established that knowledge directly influenced attitudes (e.g., perceived risk and efficacy beliefs) and practices [9]. However, a study by Islam conducted in Bangladesh established insufficient knowledge but more positive attitudes toward the COVID-19 vaccine among the general population. Therefore, he recommended an instant health education program to be initiated so as to enhance mass COVID-19 vaccination [14]. Thus, the primary objective of this study was to assess knowledge, attitudes, and practices toward COVID-19 vaccination and its associated factors among the general adult population in Malawi so as to guide the policy direction in combating the COVID-19 pandemic in Malawi.

## 2. Methodology

### 2.1. Study Design

This study employed a population-based cross-sectional study design as it aimed at capturing current individual knowledge, attitudes, and practices on COVID-19 vaccinations at a single point in time to enlighten related stakeholders like policymakers and practitioners.

### 2.2. Study Setting and Period

The study was conducted in 16 selected districts from the three regions of Malawi, namely, northern, central, and southern. The districts within the regions of Malawi were selected based on the level of SARS-CoV-2 burden in the area, as this served as a guide in health resource allocation and interventions (Appendix A). The study data was collected in the months of June and July 2022.

### 2.3. Target Population

The study targeted adult respondents (aged 18 years and above) of the male and female genders residing in rural and urban communities of selected districts from the northern, central, and southern regions of Malawi [8,9,10,11,12].

### 2.4. Sampling Technique and Tools

The study employed a multi-stage sampling technique to select participants from each district who satisfied the inclusion criteria. At the regional level, districts were stratified into high and low SARS-CoV-2 burden areas. The COVID-19 disease burden of the district was calculated as the proportion of the total number of confirmed COVID-19 cases to the total district population (Appendix A) [15,16]. Districts falling into the two strata (low and high) of the SARS-CoV-2 burden were purposefully selected from each region.

Four representative districts were selected from the northern region of Malawi, namely Mzimba and Likoma, which were high COVID-19 burden areas, while Chitipa and Nkhatabay were low COVID-19 burden areas. In the central region of Malawi, Lilongwe, Ntchisi, and Salima were selected as high COVID-19 burden areas, while Ntcheu and Dedza were selected as low COVID-19 burden areas. In the southern region of Malawi, Blantyre, Neno, and Zomba were selected as high COVID-19 burden areas, while Machinga, Thyolo, Chikwawa, and Nsanje were selected as low COVID-19 burden areas. The number of participants to be sampled in each district was calculated using the following formula:$$\frac{Disease\;burden\;for\;the\;district}{Sum\;total\;of\;all\;the\;16\;district\;disease\;burden}\times the\;total\;sample\;size$$

Details of the total number of participants sampled in every district are provided in Appendix A, while the number of enumeration areas and households per district is provided in Appendix A.

Census Enumeration Areas within the selected district, as defined for the 2018 Malawi Census, served as primary sampling units (see Appendix A). The enumeration area (EA) was defined as the smallest functional area with clearly defined boundaries on the map, in tandem with the workload given to a single census enumerator. On average, an enumeration area had 215 households. The enumeration areas falling within the sampling frame of the selected district within the regions of Malawi were geographically stratified by urban/rural code, traditional authority, and EA codes in line with a probability proportionate to size [17].

A sampling frame comprising a list of households in the enumeration area was created. Systematic random sampling was employed to select households. One eligible participant as a unit of analysis was randomly selected within each household. We used a structured questionnaire, which was translated from English to major local languages such as Chichewa and Chitumbuka to capture all required data. We piloted 2% of the targeted sample size among individuals outside the target population. After piloting the questionnaire, we modified it and made it fit for the study. The questionnaire sections consisted of demographic characteristics, questions on knowledge (awareness of the availability, types, side effects, who is eligible, etc.), questions on attitudes (reliability, safety, effectiveness, etc.), and questions on practices (action taken to get vaccinated and challenges faced).

### 2.5. Sample Size Calculation

The sample size was determined using the single population proportion formula by Kish, considering the following statistical assumptions: 95% confidence level, 0.73 from a pooled global KAPS on COVID-19, and 2% margin of error [14,15,18]. The following single-proportion formula was used:n=zα22p1−pd2
where *n* = initial sample size, *z* = 1.96, the corresponding *z*-score for the 95% CI, *p* = proportion (73%), and *d* =margin of error (0.02). The initial sample size for the study was 1893 and was adjusted for a 10% non-response rate and a design effect of 1.5, giving the final sample size of 3155.

### 2.6. Data Quality Assurance

Well-trained research assistants who were fluent in the local languages and familiar with the local customs were recruited to collect the data. Investigators and supervisors strictly monitored the data collection process. The data collected were securely kept at the Public Health Institute of Malawi (PHIM).

### 2.7. Data Analysis

Statistical Package for the Social Sciences (SPSS) was used for data cleaning, editing, sorting, and coding. Descriptive statistics (i.e., frequencies, percentages, means, and standard deviations) and first-order analysis (i.e., chi-square tests and Fisher’s exact tests) were performed using SPSS Version 25 (Chicago, IL, USA). A Pearson chi-square test was performed to determine the association of equally weighted above-median composite binary variables from knowledge, attitudes, and practices scores on COVID-19 vaccination with socio-demographic information. All statistical tests were considered significant at a 95% confidence interval with a *p*-value of less than 0.05.

To calculate the composite scores on knowledge, seven questions with responses coded as No or Yes, including the following: Do you know about the COVID-19 vaccine; Which COVID-19 vaccines are currently being administered in Malawi; Is it dangerous to use vaccine overdose; Does vaccination increase allergic reactions; Children do not require to get the COVID-19 vaccine; And a vaccine against COVID-19 is available free of charge, were used to assess knowledge. One point was assigned for each correct response, yielding a COVID-19 knowledge composite score of 0 to 7. Individuals who scored at or above the median on the COVID-19 vaccine knowledge items were classified as having good knowledge; those who scored below the median were classified as having poor knowledge [19]. To calculate the composite scores on attitude, five questions were rated using a three-point Likert scale: undecided (0), disagree (1), agree (2), including “do not rely on the COVID-19 vaccine due to emergency development during a pandemic; vaccination is not required because immunity will be acquired naturally by infection; the newly discovered COVID-19 vaccine is safe; it is not possible to reduce the incidence of COVID-19 without vaccination; and in favor of the vaccine against COVID-19 because it is effective and safe”. Respondents who scored above the median on the COVID-19 vaccine attitude items were classified as having a favorable attitude toward the COVID-19 vaccination, and those who scored otherwise were classified as having a not favorable attitude [20].

### 2.8. Ethical Consideration

Ethical clearance for the study was obtained from the NHSRC (approval number: Protocol #22/03/2888). An approval letter to conduct the study was obtained from the Principal Secretary for Health. Permission to conduct the study in each village was sought from the traditional authorities (TA) or village headmen, and written consent was obtained from each participant before administering the questionnaire. The questionnaire was coded instead of using names as identification markers, and hence, confidentiality was ensured throughout the study.

## 3. Results

### 3.1. Socio-Demographic Characteristics of Study Participants

A total of 3068 individuals participated in the study (Table 1). Among these, 1947 (63.6%) were female and 1116 (36.4%) were male. The majority of the study participants were in the age group of 18–60 (92%). About 1039 (34.1%) participants had incomplete primary education; 626 (20.6%) did not complete their secondary education; 527 (17.3%) and 359 (11.8%) completed secondary and primary education, respectively; 326 (10.7%) did not have any educational background; and 169 (5.5%) had at least a college education background. The participants were from 16 different districts in Malawi, with 2046 (66.7%) from urban settings and 1022 (33.3%) from rural locations. Among these, 2755 (90.7%) belonged to the Christian religion. Notably, 1498 (49.9%) were unemployed, while 1322 (43.9%) were businesspersons, and 85 (6.2%) were employed. Additionally, most of the respondents (2086; 69.3%) were married. Worth noting was that 1816 (60.3%) respondents’ monthly household income was less than MK35,000, followed by 534 (17.7%) in the range of MK 35,000 to MK 50,000, 370 (12.3%) within MK 51,000 to MK100,000, 267 (8.9%) within MK100,000 to MK500,000, and lastly 24 (0.8%) earning over MK500,000 per month. Most of the participants did not have chronic illnesses (85.9%); however, 52 (1.7%) reported a history of respiratory and lung disease, and 152 (5%) had HIV/AIDS.

### 3.2. Knowledge, Attitudes, and Practices of COVID-19 Vaccination

Remarkably, 2936 (95.7%) knew about the COVID-19 vaccine, and 2063 (68.4%) were aware that the vaccine is effective (Table 2). Knowledge on which COVID-19 vaccine was being administered in Malawi varied among respondents, with 1639 (53.4%), 1840 (60.0%), and 817 (26.6%) indicating it was AstraZeneca, Johnson & Johnson, and Pfizer, respectively (Table 1). About 1006 (33.5%) had perceived knowledge that vaccination increases allergic reactions, while only 765 (26.4%) professed that vaccination increases autoimmune diseases (Table 2). Additionally, 1228 (41.5%) indicated that children do not require COVID-19 vaccination, and 2736 (91.3%) were aware that vaccination against COVID-19 was available for free (Table 2).

Almost 49.72% were of the view that the COVID-19 vaccine is reliable, although it was developed hastily; however, 31.75% thought it was not proper to rely on such a type of vaccine (Figure 1A). Nearly 64.48% indicated that vaccination was not necessary as one could become immune after infection (Figure 1B). On the contrary, 23.9% thought irrespective of natural immunity, vaccination was still necessary (Figure 1B). Approximately 69.29% of respondents thought the COVID-19 vaccine was safe, whereas 19.42% thought the vaccine was unsafe, and 11.34% were unsure of the safety of the vaccine (Figure 1C). Around 57.1% of study participants thought COVID-19 vaccination was critical in reducing the incidence of COVID-19, but 29.74% indicated that the incidence of the disease could decrease without vaccination (Figure 1D). Largely, 70.2% indicated that COVID-19 vaccines were safe and they would undoubtedly get vaccinated (Figure 1E). Nevertheless, 12.55% questioned the safety of the vaccine and expressed their hesitance to get vaccinated, whereas 17.23% were not sure whether the vaccine was safe or if they would get vaccinated (Figure 1E).

Concerning vaccination practices, only 1180 (38.7%) got vaccinated, while 1796 (58.9%) never got vaccinated, and 43 (1.4%) went to vaccination centers but were not offered the service (Figure 2).

### 3.3. Factors Influencing Knowledge, Attitudes, and Practices towards Coronavirus Disease 19 Vaccination

Knowledge of the COVID-19 vaccination was significantly associated (*p* < 0.05) with participants’ education levels, location, occupation, marital status, household family income, and whether they were suffering from chronic illness or not (Table 3). It was also observed that there was a significant (*p* < 0.05) relationship between the respondents’ attitudes towards COVID-19 vaccination and denomination. Furthermore, there was a significant relationship (*p* < 0.05) between the rate of vaccination and the respondents’ gender, age, level of education, region, location, occupation, marital status, family size, monthly income, and whether they had a chronic illness or not (Table 3).

## 4. Discussion

The present study aimed at assessing the current knowledge, attitudes, and practices (KAP) of Malawians towards the COVID-19 vaccination. The study involved 3068 individuals from 16 districts with varying adult age groups, genders, levels of education, location, denomination, marital status, health condition, levels of monthly income, knowledge, attitudes, and practices about COVID-19 vaccination.

Unlike most studies conducted elsewhere in Africa, participants in this study showed a high (97.2%) level of knowledge regarding the COVID-19 vaccine, with 68.4% having a positive attitude towards the COVID-19 vaccine as being effective [9,21,22,23]. It is more likely that our study participants could have a better level of knowledge compared to previous studies since vaccination campaigns have been on different platforms for over a year now with massive campaigns through different media outlets. For instance, in a study that was conducted in South Korea before the COVID-19 mass vaccination began, the level of knowledge was very low [14]. This is an encouraging finding, as it shows the public is very aware of the vaccine and has a relatively better attitude. However, just like reports from other studies [8,23,24], it is worrisome that some misconceptions about the vaccine exist. Likely, this is why only about 38.7% indicated having been vaccinated. However, this reported vaccination rate was still high as compared to the vaccination rate indicated on the MoH COVID-19 Dashboard (8.3%). This likely could be due to the fact that our study mainly targeted adults, while the MoH online shows a percentage against the total national population [4]. It is also likely to be biased due to self-reporting on vaccine status, where the majority may prefer to report what is expected of them, unlike their actual status.

This study has discovered that knowledge of the COVID-19 vaccination was significantly associated with participants’ level of education, location, occupation, marital status, denomination, household family income, and whether they were suffering from chronic illness or not (*p* < 0.05).

These findings, particularly for education levels and gender, are very similar to other studies conducted elsewhere [9,14,24,25,26]. This could be due to the fact that being a pandemic, most COVID-19 information is still in international languages like English and mostly spread through social media, posing a big challenge for non-educated people and the female gender in our context.

As reported in other studies [9,14], the decision to receive the COVID-19 vaccine is a controversial aspect with a critical diversity in most epidemiological and demographic characteristics. This is why it has been observed that despite a good number of people having insignificant differences in terms of knowledge and attitudes towards vaccinations, a significant relationship was observed between the rate of vaccination and the respondent’s gender, age, level of education, region, location, occupation, marital status, family size, monthly income, and whether they had a chronic illness or not. This may require detailed unpacking of these differences so as to tailor individualized group-centered messages, which can help increase vaccine uptake. For instance, a message targeting a less or uneducated population needs to be well translated into local languages that can be easily understood by them. For instance, a multinational Delphi consensus to end the COVID-19 public health threat study suggests the need for the messages to clearly explain the efficacy and limitations of all the available vaccines in preventing SARS-CoV-2 transmission and reducing the severity of COVID-19 to the general public [27]. In addition, they recommend the engagement of trusted local leaders and organizations in vaccination efforts, especially in settings where individuals have lower levels of trust in government. It is, therefore, important to unpack the details of these findings and establish the potential individual concerns that may be precursors to vaccine hesitancy. As such, tailored messages that address the underlying bases of an individual’s concerns should be used in targeted public health communications [27].

Furthermore, there is a need to devise a mechanism to reach out to the masses, especially those who do not have access to social media. There is also a need to incorporate the vaccine messages into local village meetings and gatherings. Some of these approaches have been proposed and used in other settings [14]. As noted in other studies, there is a need to make an effort to develop strategies that can positively align with the attitude of the public towards vaccines since negative perceptions about COVID-19 vaccines are associated with poor rates of COVID-19 vaccination [15,28]. Lazarus, J.V (2022) [27] suggested that community leaders, scientific experts, and public health authorities should collaborate to develop public health messages that build and enhance individual and community trust and use the preferred means of access and communication for different populations. Furthermore, in coming up with the messages, public health professionals and authorities should combat false information proactively based on clear, direct, culturally responsive messaging that is free of unnecessary scientific jargon. These recommendations are in line with our study findings, which have shown that vaccination uptake is severely hampered due to several factors that cannot be easily achieved using foreign policies alone but locally tailored approaches.

## 5. Conclusions

Overall, the level of knowledge and attitudes about the COVID-19 vaccination was good. This study has also established that different population groups have statistically different levels of knowledge and attitudes regarding COVID-19 vaccination. Furthermore, this study has also indicated a significant relationship between the rate of vaccination and the respondent’s gender, age, level of education, region, location, occupation, marital status, family size, monthly income, and whether they had a chronic illness or not.

## 6. Recommendations

The government of Malawi should keep on promoting the awareness and attitudes campaign on COVID-19 vaccinations. This report may be used when designing public communications interventions since it is the first of its kind and gives a comprehensive available level of COVID-19 KAP on vaccination, hence the need for public dissemination of this report. COVID-19 Vaccine communication strategies are required and need to be emphasized to increase vaccine acceptability. There is a need to tailor the COVID-19 vaccine message which can focus on particular groups; for instance, those with little or no education need to have locally tailored messages in local languages. Future research may focus on longitudinal studies and comparative analyses to build upon these findings.

## Figures and Tables

**Figure 1 vaccines-12-00221-f001:**
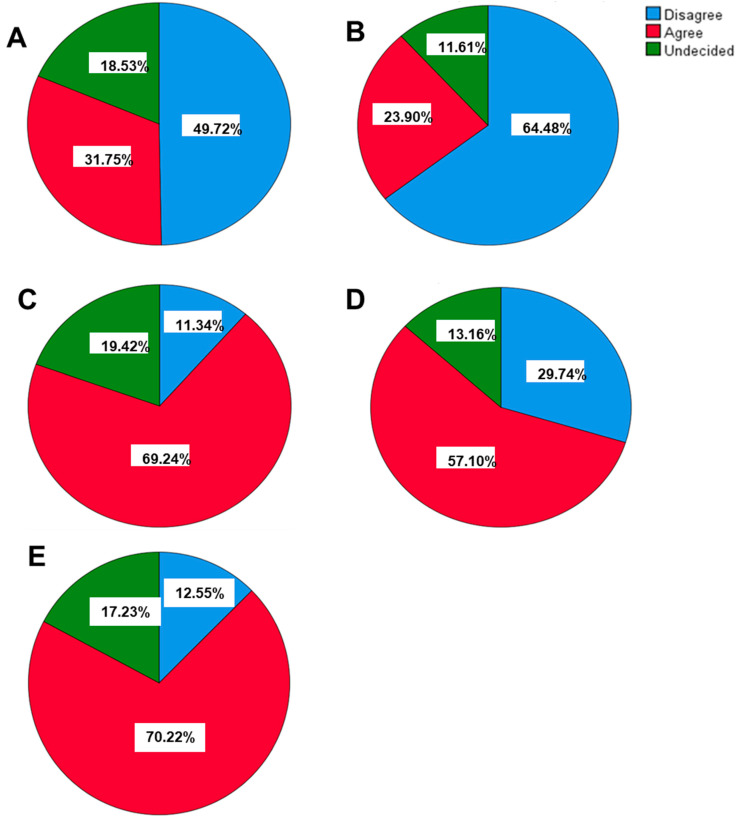
Attitudes of participants towards COVID-19 vaccine. (**A**) Do not rely on the COVID-19 vaccine due to emergency developments during the pandemic. (**B**) Vaccination is not required because immunity will be acquired naturally by infection. (**C**) The newly discovered COVID-19 vaccine is safe. (**D**) It is not possible to reduce the incidence of COVID-19 without vaccination. (**E**) COVID-19 Vaccines are safe, and I will take the COVID-19 vaccine without any hesitation.

**Figure 2 vaccines-12-00221-f002:**
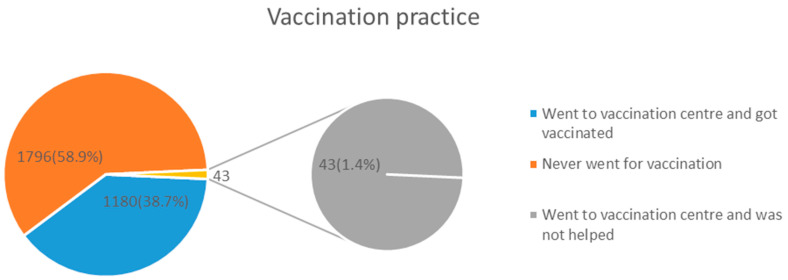
COVID-19 vaccination practices among study respondents.

**Table 1 vaccines-12-00221-t001:** Socio-Demographic Factors.

Category	Number	Percentage
Gender		
Male	1116	36.4
Female	1947	63.6
Age		
18–60	2788	92.0
61–100	242	8.0
Education		
No education	326	10.7
Primary incomplete	1039	34.1
Primary complete	359	11.8
Secondary incomplete	626	20.6
Secondary complete	527	17.3
College and above	169	5.5
Region		
Southern	1528	49.8
Central	737	24.0
Northern	803	26.2
Location		
Urban	2046	66.7
Rural	1022	33.3
Denomination		
Christian	2755	90.7
Muslim	263	8.7
Others	18	0.6
Occupation		
Unemployed	1498	49.9
Business	1322	43.9
Employed	185	6.2
Marital status		
Married	2086	69.3
Not married	566	18.8
Widowed	212	7.0
Divorced	145	4.8
Family size		
<four members	1162	39.9
>four members	1746	60.0
Monthly income of the household		
Less than MK 35,000	1816	60.3
MK 35,000 to MK 50,000	534	17.7
MK 51,000 to MK100,000	370	12.3
MK100,000 to MK500,000	267	8.9
Greater than MK500,000	24	0.8
Chronic illness		
Yes	432	14.1
No	2636	85.9
Type of chronic illness?		
Respiratory and lung disease	52	1.7
Cardiovascular disease	186	6.1
Diabetes Mellitus	42	1.4
HIV/AIDS	152	5

**Table 2 vaccines-12-00221-t002:** Knowledge of Vaccination.

Factor	Frequency (%)
Do you know about the COVID-19 vaccine? (*n* = 3068)	
Yes	2936 (95.7)
No	132 (4.3)
Do you know about the effectiveness of COVID-19 vaccine? (*n* = 3018)	
Yes	2063 (68.4)
No	955 (31.6)
Which COVID-19 vaccines are currently being administered in Malawi (*n* =3068)	
AstraZeneca	1639 (53.4)
Johnson & Johnson	1840 (60.0)
Pfizer	817 (26.6)
Not sure	807 (26.3)
Does vaccination increase allergic reactions? (*n* = 3003)	
Yes	1006 (33.5)
No	1255 (41.8)
I don’t know	742 (24.7)
Does vaccination increase autoimmune diseases? (*n* = 2894)	
Yes	765 (26.4)
No	1271 (43.9)
I don’t know	858 (29.6)
Children do not require to get COVID-19 vaccination (*n* = 2962)	
Yes	1228 (41.5)
No	1340 (45.2)
I don’t know	394 (13.3)
Vaccine against COVID-19 is available free of cost (*n* = 2998)	
Yes	2736 (91.3)
No	123 (4.1)
I don’t know	139 (4.6)

**Table 3 vaccines-12-00221-t003:** Association of COVID-19 vaccination knowledge, attitudes, and practices with socio-demographic and health-related factors.

Variables	Vaccination Knowledge		*p*-Value	Attitude		*p*-Value	Vaccination Practice		*p*-Value
	Good	Poor		Favorable	Unfavorable		Went for Vaccination	Did Not Go for Vaccination	
**Gender**									
Male	882 (87.3)	130 (12.8)	0.20	912 (92.3)	76 (7.7)	0.02	480 (43.8)	615 (51.2)	0.005
Female	1491 (85.5)	253 (14.5)		1522 (89.6)	177 (11.4)		742 (38.6)	1178 (61.4)	
**Age**									
18–60	2166 (86.2)	346 (13.8)	0.50	2216 (90.6)	230 (9.4)	0.71	1089 (39.7)	1655 (60.3)	0.000
61–100	186 (84.5)	34 (15.5)		194 (89.8)	22 (10.2)		122 (51.3)	116 (49.3)	
**Education**									
No education	247 (83.2)	50 (16.8)		258 (88.7)	33 (11.3)		134 (41.4)	190 (58.6)	
Primary incomplete	741 (81.1)	173 (18.9)		809 (89.6)	94 (10.4)		368 (36.3)	646 (63.7)	
Primary complete	282 (85.5)	48 (14.5)	0.000	294 (92.2)	25 (7.8)	0.21	146 (40.9)	211 (59.1)	0.000
Secondary incomplete	498 (88.3)	66 (11.7)		509 (92.7)	40 (7.3)		247 (40.1)	369 (59.9)	
Secondary complete	443 (93.1)	33 (6.9)		415 (91.0)	41 (9.0)		216 (41.4)	305 (58.4)	
College and above	150 (93.8)	10 (6.2)		137 (88.4)	18 (11.6)		101 (60.8)	65 (39.2)	
**Region**									
Southern	1128 (87.1)	167 (12.9)		1130 (98.9)	122 (1.1)	0.52	493 (32.8)	1010 (67.2)	
Central	583 (86.4)	92 (13.6)	0.20	601 (90.0)	67 (10.0)		358 (49.7)	363 (50.3)	0.000
Northern	665 (84.3)	124 (15.7)		705 (91.6)	65 (8.4)		372 (46.8)	423 (53.2)	
**Location**									
Urban	1594 (87.6)	225 (12.4)	0.001	1598 (90.3)	172 (9.7)	0.50	845 (41.9)	1174 (58.1)	0.03
Rural	782 (83.2)	158 (16.8)		838 (91.1)	82 (8.9)		378 (37.8)	622 (62.2)	
**Denomination**									
Christian	2141 (86.5)	335 (13.5)		2193 (91.0)	217 (9.0)	0.02	1115 (41.1)	1597 (58.9)	0.20
Muslim	201 (82.7)	42 (17.3)	0.08	208 (87.0)	31 (13.0)		95 (36.5)	165 (63.5)	
Others	10 (71.4)	4 (28.6)		13 (76.5)	4 (23.5)		5 (29.4)	12 (70.6)	
**Occupation**									
Unemployed	1124 (83.9)	215 (16.1)		1181 (90.3)	127 (9.7)		563 (38.2)	910 (61.8)	
Business	1036 (86.4)	163 (13.6)	0.000	1059 (90.8)	107 (9.2)	0.31	531 (40.8)	772 (59.2)	0.000
Employed	166 (98.2)	3 (1.8)		155 (93.9)	10 (6.1)		106 (58.0)	77 (42.0)	
**Marital status**									
Married	1599 (85.8)	264 (14.2)		1642 (90.4)	174 (9.6)		831 (40.5)	1221 (59.5)	
Not married	464 (89.4)	55 (10.6)	0.009	461 (91.5)	43 (8.5)	0.85	207 (37.2)	349 (62.8)	0.007
Widowed	153 (79.7)	39 (20.3)		167 (90.3)	19 (9.7)		108 (50.9)	104 (49.1)	
Divorced	116 (87.2)	17 (12.8)		121 (89.6)	14 (10.4)		60 (41.4)	85 (58.6)	

## Data Availability

The data for the study are available from the corresponding author upon reasonable request.

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
