# Peer review of "Knowledge, Attitudes, and Practices on COVID-19 Vaccination among General Adult Population in Malawi: A Countrywide Cross-Sectional Survey"

_vaccines, 2024, doi:10.3390/vaccines12030221_

Round 1
Reviewer 1 Report
Comments and Suggestions for Authors
The manuscript entitled "Knowledge, Attitudes and Practices on COVID-19 Vaccination among General Population in Malawi. A Country Wide Cross-Sectional Survey" is a potentially useful contribution, that could inform the public health practice in Malawi. The main weaknesses of the manuscript is the poor argumentation (difficult to understand), inconsistent reporting of methods and results and conclusions not based on provided results. See more specific comments below:
Major suggestions:
1. It is not very clear why the authors oversampled regions with high disease burden. If the authors would want to compare the knowledge and attitudes between regions more and less affected, why oversample areas with higher number of cases (especially that the higher number of cases could simply reflect better access to testing)? Not everything is clearly described. First, how is defined "burden" in the formula in line 111? Second, the results do not include any comparison of studied indicators between "high burden" and "low burden" areas. So, this effort to stratify by COVID-19 burden is a bit puzzling...
2. The authors should explain in the methods how they calculated the composite indicators "vaccination knowledge" (good/poor) and "attitutude" (favourable/unfavourable)? These seem to be important in their overall message/conclusion and need to be precisely introduced.
3. The authors should be more clear that the entire investigation is in adult population (including title, abstract, description of the study population, inclusion criteria, etc...
4. The manuscript should be proofread and improved for English language. The authors use long sentences that are sometimes difficult to follow. Also, the authors frequently use qualifiers (interestingly, notably, surprisingly), which includes their personal interpretation of the findings which is not appropriate in a scientific article.
Specific/detailed suggestions:
5. Abstract: The abstract should be improved. Sentences are long, complex, and difficult to follow. In lines 21-22, the sentence is not finished... Overall, there is too much emphasis on the background, and not enough on the results and conclusions. Because of lack of reported results, it's not clear what is the conclusion from this investigation... In lines 26-27, the authors write that "attitude was not generally good", and it is not defined what is meant by "good attitude"...
6. Introduction lines 37-40: Suggest to skip the global figures on COVID-19 burden. It's a common knowledge that the disease was widespread and affected all countries. I suggest to focus the background information on the Malawi COVID-19 experience and introducing to the reader how a KAP study can add to the Public Health practice in Malawi.
7. Introduction lines 57-60: The authors should avoid qualifiers like "Interestingly". In fact, it is completely normal and expected that people have varying level of knowledge and different attitudes towards just any medical issue/problem, so it's not surprising at all, especially when looking at various countries and regions.
8. Methods: The authors should provide more details on the content of the questionnaire, categorisation of answers and data analysis. For example if they used Likert scales, how they categorised their results? How were the composite indicators calculated?
9. Results Figure 2 lines 210-213: The caption for chart (E) is missing.
10. Discussion lines 229-237: This justification is actually more appropriate in the introduction section, as it provides some argumentation and justification how a KAP study results can help in the pandemic response. I suggest to move it the background, replacing the description of the global COVID-19 figures which are not very useful.
11. Recommendations lines 285-298: The recommendations should be placed after the conclusions and should be aligned with the conclusions (each conclusion should lead to one or more recommendations.
12. Limitations lines 299-303: There is many limitations not mentioned (see above), so the section should be updated. On the other hand, the limitations mentioned currently are not based on provided results, since the authors did not summarize the recruitment process in detail.
13. Conclusions lines 305-310: The conclusions listed are not very well backed by the presented results. The authors write that the knowledge was overall good but later they recommend many educational activities... Later they conclude about the "attitudes towards COVID testing" while there were no findings reported on COVID-19 testing at all... How did the authors reach such conclusions without any results?
Comments on the Quality of English LanguageThe English should be proofread and improved. I suggest to use precise wording, short clear sentences, and avoiding qualifiers.
Author Response
The responses have been attached

Reviewer 2 Report
Comments and Suggestions for Authors
Authors describe the design, data collection, and outcomes of a survey on COVID-19 vaccination sentiment in Malawi. These outcomes are a valuable contribution to the understanding and prioritization of public health communication, and ought to be published.
The current version of the manuscript would benefit from additional editing and improved clarity, but does not have any issues with scientific content.
Notes:
Line 21-22: Incomplete sentence in the abstract. "Among the total 3068 participants, 1947 21 (63.6%) were." Should read that 63.6% were female.
Lines 45-51: Comparing case fatality ratios between countries is complicated by differences in surveillance systems. Higher case fatality percentages in Malawi may be due to fewer mild/non-fatal cases being recorded. When mortality is reported on a per-100k population basis, COVID-19 mortality in Malawi (14) was lower than Zambia (22), or Zimbabwe (38). (Ref: https://coronavirus.jhu.edu/data/mortality). It is worthwhile citing both metrics.
Lines 210-213: Legend in Figure 2 is missing a description of 2E. The description of 2B here does not match the text in section 3.2 (lines 193-195). Did the majority of respondents think immunity after infection was sufficient?
Comments on the Quality of English LanguageMinor concerns with word choice and phrasing. Sentences such as (line 188/189) "do not require to get ... vaccination" should be "do not require ... vaccination." Manuscript would benefit from additional editing, but is understandable in it's current form.
Author Response
The responses have been attached

Reviewer 3 Report
Comments and Suggestions for Authors
The paper aims to assess the knowledge, attitudes, and practices (KAP) towards COVID-19 vaccination in Malawi. The study covers a wide demographic, including various districts and a large sample size, providing a robust dataset for analysis. Focusing on COVID-19 vaccination in a country like Malawi is both topical and critical for public health. Using a multi-stage sampling technique and structured questionnaires translated into local languages adds to the study's reliability. Using SPSS for data analysis and including various socio-demographic factors provide depth to the findings.
Drawbacks:
1. The paper acknowledges challenges in random sampling due to urbanization and natural disasters, which could impact the representativeness of the data.
2. There is a possibility of bias in self-reported vaccination status, as participants might report what is socially desirable rather than their actual status.
3. While the paper presents the data effectively, it could benefit from a deeper exploration of the implications of these findings in the broader context of global health.
4. The study does not compare its findings with similar studies in other regions or countries, which could have provided a more comprehensive understanding of the KAP towards COVID-19 vaccination.
Recommendations:
1. Develop targeted communication strategies that cater to the specific needs of different demographic groups, especially those with lower educational levels.
2. Encouraging policymakers to incorporate these findings into broader COVID-19 preventive measures and vaccination strategies.
3. Implementing vaccine risk communication strategies to address and reduce misconceptions about COVID-19 vaccination.
4. It is recommended to discuss the multinational consensus on COVID-19 in the context of the study (doi: 10.1038/s41586-022-05398-2)
The paper provides valuable insights into the knowledge, attitudes, and practices regarding COVID-19 vaccination in Malawi. Despite methodological limitations, it contributes significantly to understanding public perceptions and can guide policy and health communication strategies in Malawi and similar contexts. Future research should focus on longitudinal studies and comparative analyses to build upon these findings.
Author Response
The responses have been attached

Round 2
Reviewer 1 Report
Comments and Suggestions for Authors
The revised manuscript "Knowledge, Attitudes and Practices on COVID-19 Vaccination among adult General Population in Malawi. A Country Wide Cross-Sectional Survey" has been considerably improved compared to the original submission. However, some issues remain to be addressed to improve the manuscript.
Main suggestions:
- The abstract needs to be imrpoved. Specifically, the authors should make an effort to provide a clear conclusion and recommendations based on the provided results. Therefore, I suggest to prioritize results that are essential for their message (main conclusion). For example, the sentence "Overall, the level of attitude toward the COVID-19 vaccination was not generally good." is a bit ambigous since it is oddly phrased and it is unclear if it is a result or a conclusion. In the manuscript the authors seemed to conclude that knowledge was good and the attidude was not favourable which was reflected by the practice of not taking the vaccine. If these are the conclusions they should be included in the abstract as well and backed by relevant results. The abstract is the most read part of the manuscript and it has to be well written!
2. The authors should make sure that tables 1-3 are in the correct format. They seem to be formatted not according to the required guidelines.
3. Discussion lines 295-320: This argumentation is not based on the authors results. It is interesting and could be used in a report directed to the national authorities, but is not appropriate in this manuscript. I think the authors should stick to their findings.
4. Conclusions line 323: The authors say that "there is still a need to improve the population's attitude towards COVID-19 vaccination", which is not really a conclusion but more a recommendation. Moreover, the Table 3 reports that 92.3% of males and 89.6% females had favourable attitude. So if the attitude was so favourable why does it need improvement? I recommend that the authors revise the conclusions section and state correct conclusions based on their data (knowledge "good or poor", attitude "favourable or unfavourable" and practice) and based on their conclusions state feasible recommendations... This same should apply to the abstract - the conclusions and recommendations should be reflected there (but in a more concise way).
Comments on the Quality of English LanguageThe English level has been improved. However, the argumentation is a bit problematic in some sections.
